# Enhancing Safety and Quality of Cardiopulmonary Resuscitation During Coronavirus Pandemic

**DOI:** 10.3390/jcm14124145

**Published:** 2025-06-11

**Authors:** Diána Pálok, Barbara Kiss, László Gergely Élő, Ágnes Dósa, László Zubek, Gábor Élő

**Affiliations:** 1Doctoral College, Semmelweis University, Üllői Street 26, H-1085 Budapest, Hungary; drkissbarbara@gmail.com; 2Department of Anesthesiology and Intensive Therapy, Semmelweis University, Üllői Street 78, H-1082 Budapest, Hungary; elo.laszlo@semmelweis.hu (L.G.É.); zubek.laszlo@semmelweis.hu (L.Z.); 3Institute of Behavioural Sciences, Department of Bioethics, Semmelweis University, Nagyvárad Square 4, H-1089 Budapest, Hungary; dosaagi@gmail.com

**Keywords:** cardiac arrest, cardiopulmonary resuscitation, DNACPR order, coronavirus pandemic

## Abstract

**Background:** Professional knowledge and experience of healthcare organization went through continuous change and development with the progression of COVID-19 pandemic waves. However, carefully developed guidelines for cardiopulmonary resuscitation (CPR) remained largely unchanged regardless of the epidemic situation, with the largest change being a more prominent bioethical approach. It would be possible to further improve the quality of CPR by systematic data collection, the facilitation of prospective studies, and further development of the methodology based on this evidence, as well as by providing information and developing provisions on interventions with expected poor outcomes, and ultimately by refusing resuscitation. **Methods**: This study involved the critical collection and analysis of literary data originating from the Web of Science and PubMed databases concerning bioethical aspects and the efficacy of CPR during the COVID-19 pandemic. **Results**: According to the current professional recommendation of the European Resuscitation Council (ERC), CPR should be initiated immediately in case of cardiac arrest in the absence of an exclusionary circumstance. One such circumstance is explicit refusal of CPR by a well-informed patient, which in practice takes the form of a prior declaration. ERC prescribes the following conjunctive conditions for do-not-attempt CPR (DNACPR) declarations: present, real, and applicable. It is recommended to take the declaration as a part of complex end-of-life planning, with the corresponding documentation available in an electronic database. The pandemic has brought significant changes in resuscitation practice at both lay and professional levels as well. Incidence of out-of-hospital resuscitation (OHCA) did not differ compared to the previous period, while cardiac deaths in public places almost halved during the epidemic (*p* < 0.001) as did the use of AEDs (*p* = 0.037). The number of resuscitations performed by bystanders and by the emergency medical service (EMS) also showed a significant decrease (*p* = 0.001), and the most important interventions (defibrillation, first adrenaline time) suffered a significant delay. Secondary survival until hospital discharge thus decreased by 50% during the pandemic period. **Conclusions**: The COVID-19 pandemic provided a significant impetus to the revision of guidelines. While detailed methodology has changed only slightly compared to the previous procedures, the DNACPR declaration regarding self-determination is mentioned in the context of complex end-of-life planning. The issue of safe environment has come to the fore for both lay and trained resuscitators. **Future Directions:** Prospective evaluation of standardized methods can further improve the patient’s autonomy and quality of life. Since clinical data are controversial, further prospective controlled studies are needed to evaluate the real hazards of aerosol-generating procedures.

## 1. Introduction

The coronavirus (COVID-19) pandemic was one of the greatest global challenges of mankind since World War II. The course and rules of our daily lives changed significantly, presenting challenges that had to be faced every day. In addition, the adequacy of healthcare services, clinical procedures, and resources has become a significant topic of everyday public discourse. With the expansion of professional knowledge and experience, the organization of healthcare services for COVID-19 patients went through continuous change and development, according to the progression of pandemic waves. At the same time, carefully developed professional recommendations for resuscitation remained largely unchanged, regardless of the epidemic situation or the increased emphasis on a bioethical approach.

Out-of-hospital cardiac arrest (OHCA) and mortality rates increased considerably during the COVID-19 pandemic compared to the same period of the previous year. A temporary rise in OHCA incidence was detected, with a drop in survival. Societal response to OHCA changed during the pandemic, presenting fewer bystander cardiopulmonary resuscitations (CPRs), longer emergency medical service (EMS) response times, and worse OHCA survival rates [1]. Significant decrease of in-hospital-cardiac-arrest (IHCA) CPRR survival rates were detected during the pandemic period as well [2].

The coronavirus pandemic induced progressive discussion on end-of-life decisions in society and among healthcare workers as well. Eminent media interest was found during the first wave of the pandemic, with an emerging number of professional publications. Media coverage of do-not-attempt cardiopulmonary resuscitation (DNACPR) orders raised questions around the value and quality of life, particularly for vulnerable individuals. Media outlets stimulated advocacy and support for human rights and autonomy as well [3]. Concerns about inappropriate use of DNACPR also emerged during the pandemic. One-sided DNACPR orders that could be issued for older people, certain racial groups, or individuals with disabilities, rather than based on individual clinical assessments, emerged. This raised fears over breaches of human rights and that DNACPR orders were used discriminatorily [4,5]. These factors sped up the scientific approach and the evolution of professional guidelines concerning DNACPR and end-of-life decisions.

The International Liaison Committee on Resuscitation (ILCOR) reached consensus on the International Resuscitation Guidelines by systematically evaluating the evidence for resuscitation standards and guidelines and identifying national and regional differences, including DNACPR orders, in the context of complex end-of-life planning. Subtle differences in regional and national resuscitation guidelines remained evident [6].

Up to this time, there is enough information on the epidemiology, societal effects, and efficacy of CPR, whereas only a few studies can be found concerning DNACPR orders and complex end-of-life planning during the coronavirus pandemic. The aim of this narrative review was to synthetize bioethical and adequacy concerns of cardiopulmonary resuscitation (CPR) considering a safe environment during coronavirus pandemic.

## 2. Materials and Methods

Critical collection and synthesis of literary data concerning bioethical aspects—especially DNAR orders—and the efficacy of CPR during the COVID-19 pandemic was performed in this narrative review by using the Web of Science and PubMed databases. The keywords used for literature search included “cardiopulmonary resuscitation”, OR “cardiac arrest” AND “coronavirus pandemic” OR “COVID-19” AND “End-of-Life decisions” OR “DNACPR” OR “DNAR”.

## 3. Results

### 3.1. Epidemiological Data on Cardiac Arrest in the Context of the Coronavirus Pandemic

European data on OHCA cases are already available in about 70% of countries and are also widely available on IHCA. The incidence of OHCA in Europe is 67–170/100,000 inhabitants, and the incidence of IHCA is 1.5–2.8 per thousand hospital admissions. A proportion of 50–60% of out-of-hospital resuscitations (19–97/100,000 people) are started or continued by trained personnel. The rate of bystander resuscitation shows significant variability by country and region (58%, 13–83%). Dispatch-assisted CPR by telephone (T-CPR) is also available in 80% of countries, and 75% of the member states also operate an automated external defibrillator (AED) register. The survival rate for OHCA shows an average of 8%, with a significant standard deviation (0–18%), which is a slight increase compared to previous data. The survival rate following in-hospital resuscitation varies between 15% and 34% regarding 30-day survival and hospital discharge [7]. Epidemiological data clearly show that the effectiveness of resuscitation is still very low. Therefore, it would be possible to further improve their quality by systematic data collection and the facilitation of prospective studies and further development of the methodology based on the evidence obtained in this way, as well as by providing information and developing consensual provisions on interventions with expected poor outcomes, and ultimately by refusing resuscitation [8].

The incidence of OHCA significantly increased in regions with higher weekly COVID-19 incidence, reaching up to 136 cases per 100,000 people per week. This rise is likely attributable to severe complications associated with SARS-CoV-2 infection—such as hypoxic respiratory failure, thromboembolic events, and myocardial injury—which may have directly or indirectly triggered cardiac arrest. Social and healthcare-related factors during the pandemic, such as limited access to emergency care, may have also played a contributing role. The proportion of OHCA of medical origin increased from 87.5% to 89.2% (*p* < 0.001), indicating that most cases were likely linked to COVID-19-related medical conditions, such as respiratory failure or sepsis. There was also a marked increase in OHCA occurring at home, from 67.4% to 74.7% (*p* < 0.001), likely due to lockdown measures and reduced public mobility. Furthermore, regional disparities were observed, with worse OHCA outcomes in high COVID-19 incidence areas, independent of previous predictors. This suggests that the pandemic’s systemic effects—such as strained healthcare capacity and delayed care—significantly impacted survival outcomes [9].

The onset of the pandemic has had considerable impact on outcomes for IHCA cases as well. A significant drop in the survival rate was recorded from 25% to 21%. Nearly 25% of patients with IHCA had suspected or confirmed COVID-19 infection, which was associated with 58% lower odds of survival by using a risk-standardized survival rate model [2].

### 3.2. Refusing CPR and Complex End-of-Life Planning

Modern bioethics and patient rights regulations in secularized countries are based on the self-determination of the competent person; therefore, the patient has the right to refuse life-saving treatment (do not attempt cardiopulmonary resuscitation—DNACPR). However, due to the critical time factor, the certainty of death if the intervention is not carried out, and the incapacity of the person experiencing cardiac arrest, consent to treatment must be assumed in the absence of the patient’s previous valid disposition or other exclusionary reason. According to the relevant professional recommendation of the European Resuscitation Council (ERC), resuscitation should also be started immediately in the absence of an exclusionary circumstance. The relevant European (ERC) recommendation prescribes the following conjunctive conditions for DNACPR declarations: present, real, and applicable. It recommends the use of complex end-of-life planning and corresponding documentation available in an electronic database [10].

The decision of patients to refuse lifesaving or life-sustaining treatment is largely based on information from the doctor. However, aspects reflecting the value judgment of the professional society may also appear in the information, which may differ significantly from the patients’ preferences. Inadequate doctor–patient communication, doctors’ values differing from those of their patients, and excessive patient expectations can all be contributing factors [11]. About a third of patients and their relatives do not agree with the treatment considered appropriate by doctors, which can degrade both the quality of treatment and patient satisfaction according to our multicenter trial [12]. Therefore, thorough and continuous communication between the patient, relatives, and medical personnel is extremely important. Ethical training for doctors can further improve the validity of the provisions (the patient’s preferences are more adequately reflected in the provisions) and thus the quality of care, according to several clinical studies, including our own [4,11,13,14,15]. One further potential complication in implementing refusal of care in a practical setting is the highly time sensitive nature of CPR being largely irreconcilable with the need to verify the patient’s prior DNACPR declaration. To this end, ERC recommends the use of purpose-designed electronic administration. The DNACPR provision can serve as a basis for refusing further treatment during end-of-life decisions as well. The relevant ERC guideline therefore recommends paying more attention to complex end-of-life planning (advance care plans), which includes communication with the patient and family members regarding relevant issues, offering end-of-life planning to high-risk patients, and accepting and supporting it if requested by the patient [10].

The COVID-19 pandemic created complex challenges regarding the timing and appropriateness of DNACPR code status orders. A cohort study conducted among hospitalized COVID-19 infected patients showed a higher prevalence of DNACPR orders that were written earlier in the hospital course for patients hospitalized early in the pandemic versus later, despite similarities in clinical characteristics and medical interventions. Changes in clinical care between cohorts may be due to fear of resource shortages and changes in knowledge about COVID-19 [16]. A secondary retrospective analysis of data extracted from digitally stored Recommended Summary Plan for Emergency Care and Treatment (ReSPECT) showed that DNACPR was more likely present for patients lacking capacity, of increasing age (notably for patients aged over 74 years), female patients, and those with multiple comorbidities. DNACPR was less likely addressed for patients having ethnicity recorded as Asian or Asian British and Black or Black British compared to White. Having a preferred place of death recorded as ‘hospital’ led to a five-fold increase in the likelihood of dying in hospital [17]. Other studies also demonstrated significant differences in the presence of DNACPR orders depending on the age, race, and disabilities among patients infected with COVID-19 [4].

There were various methods, such as ReSPECT [15,17] or advance care planning (ACP) [18], implemented into clinical practice in order to enhance patient autonomy by standardizing the decision process. Standardization of end-of-life procedures and continuous education and training of clinicians both are key factors of the quality of end-of-life care [18,19].

End-of-life planning and, as part of this, DNACPR orders are therefore based on the patient’s autonomous decision, which is supported by both the legal systems and the relevant professional recommendations. A properly informed patient, appropriate physical and mental condition, and an influence-free decision-making context are the combined conditions of an autonomous decision. It is also necessary that the DNACPR statement—preferably as part of the complex end-of-life plan—be immediately available and applicable in specific cases. Table 1. summarizes the relevant studies and their main conclusions.

### 3.3. Proper Cardiopulmonary Resuscitation During Coronavirus Pandemic

Epidemiological data show that ensuring and improving the quality of resuscitation are based on the classic methodology of quality management. The main elements and methods of this process are as follows:Systematic collection of data based on the Utstein criteria;Data evaluation and planning of modifications by national and international scientific societies;Practical introduction of the guidelines into everyday practice, in which structured education and training play an important role.

The quality and usefulness of resuscitation are determined by evidence-based professional rules. Due to the significant time constraints, the early introduction and consistent application of the chain-of-survival is extremely important. This can be achieved by providing the appropriate level of training for all workers within the healthcare facility, maintaining their skills, organizing a suitable emergency care team (MET team), accurately documenting resuscitations (Utstein), and evaluating these factors regularly. In case of out-of-hospital resuscitation (OHCA), wider, regular layperson (bystander) education in basic resuscitation supplemented with defibrillation (BLS-AED), making AED devices widely available, and maintaining a service capable of continuously providing advanced resuscitation (ALS) can ensure the appropriateness of the intervention. All of these are complemented by the protection of the person performing resuscitation (safe environment). This aspect gained special importance during the COVID-19 pandemic [6,8].

The pandemic has brought significant changes in resuscitation practice at both lay and professional levels. The incidence of out-of-hospital CPR in OHCA cases did not differ compared to the previous period, according to a study by Australian authors, while cardiac deaths in public places almost halved during the epidemic (*p* < 0.001), as did the use of AEDs (*p* = 0.037). The number of resuscitations performed by the emergency medical service (EMS) also showed a significant decrease (46.9% and 40.6%, respectively, *p* = 0.001), and the most important interventions (defibrillation, first adrenaline time) suffered a significant delay. Secondary survival until hospital discharge thus decreased by 50% during the pandemic period (*p* = 0.002). A potential explanation for this apparent drop in the number of reported cardiac deaths in public and layperson-initiated CPR is an increased reluctance to initiate CPR during the COVID-19 pandemic for fear of contacting the infection. Another potential explanation is a drop in the number of people in public places who were susceptible to sudden cardiac death. This is partly due to widespread curfews during the pandemic, which isolated those at risk from potential resuscitators, and partly because such people were already hospitalized due to COVID infection [20]. A retrospective study highlighted the impact of the COVID-19 pandemic on OHCA successful CPR rates as well. The results showed that the overall chance of ROSC halved in individuals resuscitated during the pandemic (OR 0.494) [21]. A meta-analysis of 48 studies showed significant reductions in survival to discharge and 30-day survival rates during the pandemic as well. In addition, regional analysis highlighted significant reductions in survival-to-discharge rates in the United States and Europe but not in Asia. The effects of COVID-19 infection on cardiac arrest can be categorized as either direct or indirect. Direct factors include hypoxia, septic reactions, pulmonary embolism, acute coronary syndrome, and drug-induced arrhythmias. Indirect societal factors include lockdown, home quarantine, and more frequent instances of individuals being at risk when alone, while indirect healthcare-associated factors include a reduction in emergency testing and skills, overloading of the emergency and hospital systems and consequent delays in care, the effects of wearing personal protective equipment, and a reduction in hospital staffing. These factors are related to the number of initial COVID-19 cases, and it can be assumed that the impact on the healthcare system and OHCA was lower in Asia because the number of COVID-19 was relatively low compared to Europe and North America during the initial pandemic period. The number of cardiac arrests at home significantly increased during the COVID-19 pandemic, as the EMS response times were significantly longer in all three regions during the COVID-19 pandemic [22].

The response of CPR performed by a bystander (bCPR) doubles the chances of recovery from OHCA [23]. The pandemic has appreciably influenced behavioral patterns and social structures. The rate of bystander witnessing significantly decreased during the pandemic in comparison to the period before the pandemic (49.9% vs. 55.3%). Similarly, decreases were observed in the case of public access defibrillator activation (5.2% vs. 5.7%), shockable rhythm (9.5% vs. 11.7%), time from EMS activation to arrival on scene, survival to hospital admission (9.9% vs. 16.5%), survival to hospital discharge (7.0% vs. 10.4%), as well as good neurological outcome (6.1% vs. 8.7%) [24]. However, some countries or districts experienced an increase in bCPRs during the COVID-19 pandemic. This may be due to the lockdown and movement restrictions, during which OHCA cases were more likely to occur in residential settings. In addition, the aid of bCPR was provided by the patient’s acquaintances, who were motivated to help regardless of fear of contagion [25]. In a survey on willingness and barriers performing CPR respondents were less likely to deliver CPR to strangers than to family members or acquaintances because of reported fear of the SARS-CoV-2 pandemic [26].

In a meta-analysis of six similar studies, the authors also concluded that both certified cardiac arrests and CPR cases initiated by laypersons and emergency services (EMS) decreased during the pandemic period, especially in Europe. In addition, ambulance response times were significantly delayed in all studies. A total of 4–6% of the examined patients were SARS-CoV-2 positive and 5–26% had clinical symptoms (fever and cough or dyspnea) in 2020. Short-term consequences of OHCAs were worse during the pandemic than during the non-pandemic period, which may have been caused directly by the coronavirus infection and the indirect effects of the closure and disruption of health systems [27]. A recent cohort study showed that the coronavirus pandemic had considerable impact on the EMS management of OHCA as well, including fewer intubation attempts, less vascular access, more cases pronounced dead in the out-of-hospital setting, and time delays in almost all key OHCA interventions when compared with the pre-pandemic period. The workforce entered the pandemic with minimal experience in wearing high levels of PPE, which subsequently contributed to delays in providing interventions for patients with OHCA. A brief delay in the initiation of chest compressions can significantly decrease the survival [28]. At the same time, a systematic review based mainly on simulation studies did not prove any association between wearing PPE and reduced CPR quality or lower cardiac arrest survival. However, rescuers wearing PPE may report more fatigue [29]. Timing and dosage of adrenaline [21,30] and antiarrhythmics may also be contributing factors to worsening outcomes [31]. Barriers to performing CPR are also related to the loss of knowledge in the field of CPR over time. It was a considerable factor during the pandemic because of the absence of skill trainings during lockdowns [26]. There were small but clinically relevant time delays in the initiation of chest compressions, team arrival, first rhythm analysis, epinephrine administration, and airway insertion following the implementation of the COVID-19 PPE (Code Blue) policy in IHCA cases as well. As per policy, AED devices were used more commonly during the COVID-19 period. There was no difference in the duration of resuscitation efforts (19 min). The introduction of the COVID-19 PPE protocol was associated with delayed processes of care but comparable CPR quality [32].

Any maneuver performed during CPR is recommended to be considered an aerosol-generating procedure. Limited data from the coronavirus epidemic suggests that the baseline risk of infection among healthcare workers may be as much as 10%. Performing invasive procedures enhances the risk of infection. Endotracheal intubation is associated with a six-fold, noninvasive ventilation with a three-fold, chest compression with a 1.5-fold, and defibrillation with a 2.5-fold higher risk of infection [33]. The WHO recommendation, therefore, identified resuscitation activity as a significant risk of infection due to the formation of aerosols, and therefore prescribed the use of personal protective equipment (PPE) for the intervention in the early stages of the pandemic. The complete protective clothing for aerosol-forming interventions consists of the following: water-repellent full-body protective clothing, rubber gloves, well-fitting eye protection, an FFP3 mask. High-risk interventions in terms of aerosol formation, in order of probability, are as follows:Tracheal intubation, extubation, and related activities;Noninvasive ventilation (NIV);Tracheostomy;Mask ventilation.

Further aerosol-forming, but less dangerous interventions are positive pressure ventilation with an open airway, open tracheal secretion suction, bronchoscopy, spitting, high-flow nasal oxygen therapy, certain dental interventions, nasogastric tube removal, and chest compressions [34]. Later studies could not conclusively verify the initial assumption regarding the aerosol generation of resuscitation, especially in the case of chest compressions and defibrillation. However, the limited nature of the evidence was emphasized [35]. In addition, a cross-sectional analysis of bystander behavioral changes highlighted changes in CPR willingness during the COVID-19 pandemic, underscoring the importance of PPE in increasing people’s willingness to perform CPR, particularly with strangers whose COVID-19 infection or vaccination status may be unknown. Participants were more concerned about droplet or airborne transmission than transmission through skin contact or fomites, suggesting that an understanding of transmission routes influenced their protective measures and readiness to intervene [36]. A recent study on the quantitative evaluation of aerosol generation during CPR confirmed that CPR in humans generates high concentrations of respiratory aerosol, and these concentrations were consistently higher than those seen in awake and anesthetized humans [37].

Table 2 summarizes the relevant studies and their main conclusions.

## 4. Discussion

The coronavirus pandemic provided a significant impetus to revise the guidelines. ILCOR and ERC have issued the following recommendations [38,39].

In case of lay resuscitators, adequate protective clothing would be unlikely to be available. However, sudden cardiac death often occurred at home, where the resuscitator present was already a contact person. In this case minimizing the number of people present, covering the face of the adult to be resuscitated, omitting the breath test, and only using chest compressions was recommended. When reviving children, supplementing chest compressions with ventilation was recommended for those lay resuscitators who were willing, trained, and able to do so. In a public place, if an AED was available, its use was recommended due to the minimal risk of infection.

In case of professional resuscitators, wearing appropriate protective equipment (PPE) was recommended. The application of mechanical resuscitation (m-CPR) was also recommended, using devices suitable for external compression (e.g., LUCAS). To reduce aerosol formation, early, professional intubation and mechanical ventilation, and the use of a HEPA filter were recommended. Reducing the number of people present as much as possible, and clearly and urgently communicating the possible risk of infection were also recommended [40].

The COVID-19 pandemic also modified the approach of first responders to OHCA. With the rise in OCHA during the pandemic in several geographic regions and the risks of disease transmission with concurrent equipment shortages, novel noninvasive, adjunct tools, such as the point-of-care ultrasound, warrant consideration [41].

In case of in-hospital resuscitation (IHCA), wearing the above-mentioned full protective clothing for resuscitation personnel was recommended. The introduction of end-of-life planning and discussions with the patient or relatives were recommended after each patient was admitted to the hospital with proven severe COVID-19 infection. It was identified that the enhanced monitoring of patients in deteriorating condition may reduce the need for emergency intubation with the highest aerosol generation. For already intubated and ventilated patients, the use of a HEPA filter, 100% oxygen, a tidal volume of 6 mL/bwkg, a respiratory rate of 10/min, deactivation of the auto trigger, and application of appropriate end-expiratory pressure (PEEP) were recommended. It is necessary to turn a prone patient into the supine position in case of resuscitation [38]. However, a narrative review has found the prone position to be feasible for CPR, which is especially relevant for prone-ventilated COVID-19 patients [42].

Based on prior communication, limited resuscitation can be considered in patients with severe and deteriorating general conditions when treatment is futile. In such cases, chest compressions are abandoned (do not initiate compressions—DNIC) and, if necessary, defibrillation takes place [39].

The guidelines can be applied to patients specifically suspected of having, or confirmed to have, COVID-19 infection. In case of uncertainty, it is always recommended to consider treatment based on a dynamic risk assessment that estimates the prevalence of coronavirus, the patient’s individual circumstances (e.g., contact status, symptoms), the probability of the treatment’s effectiveness, and the availability of personal protective equipment (PPE). These factors can change dynamically according to the epidemic situations. Chest compressions and CPR can generate aerosols, although certainty of this evidence is low. During the COVID-19 pandemic, lay resuscitators had to consider compression-only resuscitation and using public access defibrillation (AED). Medical professionals are recommended to use personal protective equipment (PPE) for aerosol-generating procedures during resuscitation on patients infected with COVID-19, and at the same time, to consider defibrillation based on a risk/benefit analysis before starting aerosol-generating activities [43]. High aerosol concentrations can imply high risk of airborne disease transmission, but additional investigation is required determine the risk of airborne disease transmission from aerosol and its environmental dispersion [37].

With a dramatical drop in both attempts and survival rates of CPR, especially in OHCA cases, the coronavirus pandemic also offered opportunities to enhance both the quality and safety of OHCA and IHCA resuscitations. Systematic reviews and meta-analyses concerning COVID-19 patients’ CPR both highlight the heterogeneity of clinical studies and emphasize the need of standardization, as well as the eminent role of continuous education and training [1,28,29]. Assessing key factors influencing successful CPR outcomes in OHCA cases gives a relevant tool for evaluating and comparing OHCA attempts [21]. A validated tool for risk-standardized survival rates (RSSRs) for IHCA, updated for COVID-19 events, enables the comparison and benchmarking of several IHCA CPR events in different hospitals and regions [2].

## 5. Conclusions

The COVID-19 pandemic had a significant negative impact on the frequency of cardiac arrest, CPR, and survival. While the detailed methodology of CPR has changed only slightly compared to the previous procedures, DNACPR declaration regarding self-determination is mentioned in the context of complex end-of-life planning, and in addition, the issue of safe environment has come to the fore for both lay and trained resuscitators.

## 6. Limitations, Future Directions

The main limitation of this review is the narrative modality of the evaluation. Relatively few articles can be accessed concerning bioethical dimensions, especially DNACPR, by using these databases which is another limitation of the review. Limited data on long-term effects and definitive cause of OHCA should be addressed in future studies to understand the relative impact of direct or indirect effects more deeply. More descriptive epidemiological data are needed concerning the clinical implication of the recommended DNACPR orders (present, real, and applicable) and complex end-of-life planning. The prospective evaluation of standardized methods can further improve the patient’s autonomy and quality of life. Further prospective studies could evaluate more accurately the place of prone CPRs in clinical practice. Anew dimension of safety in the environment has emerged, involving an excessive number of potentially contagious patients undergoing aerosol-generating procedures. Clinical data are controversial; therefore, further prospective controlled studies are needed to evaluate the real hazards of aerosol-generating procedures.

## Figures and Tables

**Table 1 jcm-14-04145-t001:** Studies on DNACPR and End-of-Life planning concerning coronavirus pandemic.

Author, Year	Design	Aim	Main Results	Conclusions
Robinson, 2023 [4]	Editorial	Reflection on concerns about inappropriate use of DNACPR during the pandemic.	Patients prefer that discussions on DNACPR happen earlier in their illness. Doctors found DNACPR discussions difficult. Training structure did not anticipate problems.	Clinicians need suitable communication skills, training, and support. More effective public health education is required to share information about advance care planning.
Chen, 2019 [11]	retrospective, observational study.	Determination of the likelihood of signing a DNR order for patients varies among individual physicians.	Each individual physician’s likelihood of signing DNR orders was significantly different from each other. Some physicians were more likely than others to issue such orders.	Individual attending physicians influenced patients’/surrogates’ do-not-resuscitate decision-making.
Wilson, 2019 [12]	multicenter, prospective, observational study	Determination of the prevalence of perceived inappropriate treatment, impact on adverse outcomes, and discordance with clinicians.	Perceived inappropriate treatment was associated with moderate or high distress for 55% of patient/surrogate and 35% of physician/nurses and was associated with lower satisfaction and trust.	There was disagreement between clinicians and patients/surrogates about the appropriateness of treatment. It was associated with prognostic discordance and lower patient/surrogate satisfaction. Patients/surrogates who reported inappropriate treatment also reported lower satisfaction and trust in the ICU team.
John, 2021 [14]	prospective, observational study	Development of an online module regarding DNACPR conversations, aimed particularly at doctors but accessible to all clinical staff.	Average module completion time was 31 min. Quantitative feedback suggested greater confidence in theory than in practice. Mean value for theory versus practice score was 4.17/5, compared to 3.93/5.	An interactive online module presented information and basic orientation about DNACPR decisions. It introduced learners to commonly encountered communication challenges and offered strategies for helping patients or relatives to understand the risks and benefits of CPR while maintaining their trust.
Hartanto, 2022 [15]	cross-sectional survey	Comparison of changes in clinicians’ knowledge, skills, and attitudes regarding ReSPECT during the pandemic.	ReSPECT telephone discussions were more challenging when conducted during the pandemic. The importance of reaching a shared understanding was reported.	There were differences in clinicians’ knowledge, skills, and attitude scores before and during the pandemic. Findings highlighted that clinicians could benefit from training in remote ReSPECT conversations with relatives.
Comer, 2023 [16]	retrospective cohort study	Determination of differences in the utilization of DNACPR orders during different time periods of the COVID-19 pandemic.	A total of 19% of all patients had a documented DNACPR order. Age and hospitalization early in the pandemic were associated with having a DNACPR order. In-hospital mortality did not differ among cohorts among patients with DNACPR orders.	There was a higher prevalence of DNACPR orders, and these orders were written earlier in the hospital course for patients hospitalized early in the pandemic. Changes in clinical care between cohorts may be due to fear of resource shortages and changes in knowledge about COVID-19.
Anik, 2024 [17]	Secondary retrospective data analysis	Assessing patient characteristics associated with the initiation, timing, and completion of emergency care and treatment planning.	DNAR was more likely to be recorded for patients lacking capacity, of increasing age, female, and with multiple comorbidities. DNAR was less likely to be recorded for patients belonging to specific ethnic groups. A preferred place of death as ‘hospital’ led to a five-fold increase in the likelihood of dying in hospital.	Variation in the initiation, timing, and completion of ReSPECT plans was identified by applying an evaluation framework. The digital storage of ReSPECT plan data presents opportunities for assessing trends in the completion of the ReSPECT planning process and for benchmarking across sites.
Jones, 2024 [18]	retrospective observational study	Evaluating clinical presentation, management, care planning and clinical decision-making, and after death care of care home residents who died due to COVID-19.	Most individuals presented with ‘typical’ COVID-19 symptoms (cough, fever); however, >50 presented with atypical symptoms. A total of 90% of patients had a record of do not attempt cardiopulmonary resuscitation (DNACPR) decision, but only 46% had documented advance care planning (ACP), and only 37% had a clearly documented treatment escalation plan.	Care home residents are at risk of sudden clinical deterioration and death. This evaluation demonstrates that although DNACPR is in place for most individuals, holistic planning for end of life (including ACP and clinical care plans covering the management of deterioration and escalation of care) is only present for a minority.
Michalowski, 2022 [19]	Expert opinion	The article addresses a cluster of legal uncertainties surrounding DNACPR decisions, in particular regarding the grounds for such decisions, and the correct procedures for the legally required consultation, including with whom to consult.	No uniform practice exists in this regard, and while some care homes and hospitals apply the ReSPECT framework, others use DNACPR forms to record decisions that are limited to the administration of CPR. The only situation in which an advance decision not to administer CPR is binding is where someone refuses to consent to CPR in an advance directive.	The analysis shows that all forms exhibit shortcomings in reflecting the legal requirements for DNACPR decisions. A number of changes are recommended to the forms, aimed at rendering DNACPR practice compliant with the law and more protective of the person’s human rights.

Abbreviations: ACP—advance care planning; CPR—cardiopulmonary resuscitation; COVID-19—Coronavirus-19; DNACPR—do not attempt cardiopulmonary resuscitation; DNR—do not resuscitate; ReSPECT—Recommended Summary Plan for Emergency Care and Treatment.

**Table 2 jcm-14-04145-t002:** Studies concerning proper CPR during the coronavirus pandemic.

Author, Year	Design	Aim	Main Results	Conclusions
Ball, 2020 [20]	retrospective cohort study	Investigating the impact of COVID-19 pandemic on incidence, characteristics, and survival from OHCA	The incidence of OHCA did not differ during the pandemic period. CPR by EMS significantly decreased. Cardiac arrests in public locations decreased during the pandemic period, as did the number of initial shocks. Survival-to-discharge decreased by 50%.	The COVID-19 pandemic period did not influence OHCA incidence but appears to have disrupted the system of care in Australia. However, this could not completely explain the reductions in survival rates.
Ichim, 2024 [21]	retrospective, single-center observational analysis	Detailed analysis to identify the main factors predicting survival after resuscitation from out-of-hospital cardiac arrest (OHCA).	Mortality rate was 68.7%, with non-shockable rhythms predominant among fatalities. Rural patients, though younger, had lower ROSC rates than urban counterparts. Logistic regression showed that lower adrenaline doses were associated with better ROSC outcomes.	The dose of adrenaline is a significant independent predictor of the likelihood of ROSC. By integrating additional factors such as rurality, the impact of the COVID-19 pandemic, and the type of initial rhythm into the analysis, a more accurate and robust multivariable model was constructed, which proved to outperform adrenaline dose alone.
Kim, 2023 [22]	systematic review and meta-analysis	Examining the most recent trends of change in epidemiological factors, prehospital factors, and outcomes for OHCA affected by the COVID-19 pandemic	Survival and favorable neurological outcome rates were significantly lower. Survival to hospitalization, ROSC, endotracheal intubation, and the use AED decreased significantly, whereas the use of a supraglottic airway device, the incidence of cardiac arrest at home, and the response time of EMS increased significantly.	The COVID-19 pandemic altered the epidemiologic characteristics, survival rates, and neurological prognosis of OHCA patients.
Krawczyk, 2025 [24]	systematic review and meta-analysis	Examining the primary outcomes of bystander CPR during the pandemic and pre-pandemic periods	The rate of bystander witnessing significantly decreased during the pandemic, as did AED activation, incidence of shockable rhythm, time from EMS activation to arrival at the scene, survival to hospital admission, survival to hospital discharge, as well as good neurological outcomes.	The COVID-19 pandemic contributed to a reduction in bystander CPR compared to the pre-pandemic period, but this difference was not statistically significant. The study highlights the importance of bystander intervention in emergency situations and the impact of a pandemic on public health response behaviors.
Fothergill, 2021 [25]	retrospective, observational study	Describing the incidence, characteristics and outcomes from OHCA in London during the first wave of the pandemic.	There was an increase in OHCAs during the pandemic, with a strong correlation between the daily number of COVID-19 cases the increase in OHCAs occurring in private locations. There was also saw an increase in bystander CPR during the pandemic, as well as fewer resuscitation attempts and longer EMS response times. Survival at 30 days post-arrest was poorer during the pandemic.	During the first wave of the COVID-19 pandemic in London, a dramatic rise in the incidence of OHCA occurred, accompanied by a significant reduction in survival. The pattern of increased incidence and mortality closely reflected the rise in confirmed COVID-19 infections in the city.
Jaskiewicz, 2024 [26]	online survey questionnaire developed by the authors	Assessing the number and types of barriers to CPR among medical students after the pandemic ended. This study was based on a survey.	The number of all barriers reported by respondents differed significantly and was higher in those reporting fear of infection. A total of 12 out of all 23 barriers were significantly more frequent in this group of respondents.	Barriers to CPR are still common among medical students, even despite a high rate of CPR training. The pandemic significantly affected both the number and frequency of barriers. Strangers and children, as potential cardiac arrest victims, deserve special attention. Efforts should be made to minimize the potentially modifiable barriers.
Scquizzato, 2020 [27]	systematic review	Investigating the direct and indirect effects of COVID-19 on out-of-hospital cardiac arrests.	Especially in Europe, bystander-witnessed cases, bCPR and CPR attempted by EMS were reduced during the pandemic. EMS response times were significantly delayed across all studies, and the number of patients presenting with non-shockable rhythms increased in two studies.	OHCA had worse short-term outcomes during the pandemic compared to the non-pandemic period, suggesting both direct effects of COVID-19 infection and indirect effects from lockdowns and disruptions to healthcare systems. Patients at high risk of deterioration should be identified outside the hospital to promptly initiate treatment and reduce fatalities.
Armour, 2023 [28]	retrospective cohort study	Describing the impact of the COVID-19 pandemic on the care of Canadian EMS clinicians to OHCA.	A reduction was observed in the number of defibrillations, the odds of intubation attempts, vascular access, and epinephrine administration, along with higher odds of CPR termination at the scene. Delays in the initiation of chest compression and epinephrine administration were observed, while the supraglottic airways were inserted earlier.	The COVID-19 pandemic was associated with substantial changes in the EMS management of OHCA. EMS leaders should consider these findings to optimize current OHCA management and prepare for future pandemics.
Chung, 2023 [29]	systematic review and meta-analysis	The SR aimed to examine whether wearing PPE during resuscitation affects patient outcomes, CPR quality and rescuer fatigue.	A total of 17 simulation-based studies and 1 clinical study were included. No difference was observed in survival when comparing enhanced and conventional PPE. A meta-analysis of 11 RCTs and 6 observational studies found no difference in CPR quality in rescuers wearing PPE. Rescuer fatigue was worse in the PPE group.	PPE was not associated with reduced CPR quality or lower cardiac arrest survival. Rescuers wearing PPE may report more fatigue. This finding was mainly derived from simulation studies, so additional clinical studies are needed.
Okubo, 2021 [30]	retrospective cohort study	To ascertain whether there is an association between timing of epinephrine administration and patient outcomes after OHCA.	Survival to hospital discharge and favorable functional status at hospital discharge were statistically significant and differed according to the timing of epinephrine administration, and the risk ratios for survival and favorable functional status decreased with delayed administration of epinephrine.	Findings of the study suggest that early epinephrine administration is associated with better survival outcomes in adult patients with shockable and non-shockable out-of-hospital cardiac arrest.
Lupton, 2023 [31]	secondary analysis of double-blind randomized controlled study	Evaluating how timing from EMS arrival on scene to drug administration affects the efficacy of amiodarone and lidocaine compared to placebo.	In the early group, patients receiving amiodarone, compared to those receiving placebo, had significantly higher rates of survival to admission, survival to discharge, and functional survival rates.	The early administration of amiodarone, particularly within 8 min, is associated with greater survival to admission, survival to discharge, and functional survival compared to placebo in patients with an initial shockable rhythm.
Vaillancourt, 2024 [32]	before–after prospective cohort study	Evaluating the impact of a COVID-19 Code Blue policy on IHCA processes of care, CPR quality metrics, and survival to hospital discharge.	There were relevant time delays in the initiation of chest compressions, team arrival, 1st rhythm analysis, 1st epinephrine, and airway insertion. Factors independently associated with survival were male sex, witnessed, shockable rhythm, hospital location, and COVID-19 period.	The COVID-19 Code Blue policy was associated with delayed processes of care but comparable CPR quality. The COVID-19 period appeared to be associated with decreased survival.
Miraglia, 2021 [33]	systematic review snapshot	Identifying and summarizing the potential risk of infection transmission associated with key interventions performed in the context of cardiac arrest.	The management of severe COVID-19 cases involves procedures such as noninvasive ventilation and endotracheal intubation that have the potential to generate respiratory aerosols. Limited data from the SARS epidemic suggests that the baseline risk of infection among healthcare workers may be 10%. Performing endotracheal intubation is associated with a 3–5 times higher risk.	Endotracheal intubation, non-invasive ventilation, endotracheal suction, and procedures performed during CPR should be treated as high-risk procedure and managed with the highest precautions at the scene, to guard against contact with both airborne and droplet particles.
Cook, 2020[34]	narrative review	Review seeks to add some clarity regarding modes of transmission of COVID-19, what PPE is recommended, when and why. It also explores where uncertainty exists.	Airborne transmission may occur if patient respiratory activity or medical procedures generate respiratory aerosols. These aerosols contain particles that may travel much longer distances (longer than 1 m) and remain airborne longer, but their infective potential is uncertain. Contact, droplet and airborne transmission are each relevant during airway maneuvers.	Overall, there is evidence that the use of PPE does reduce rates of disease transmission and protects staff. It is essential that staff understand the purpose of PPE and its role as part of a system to reduce disease transmission from patients to staff and other patients.
Couper, 2020 [35]	systematic review	The aim of this review was to identify the potential risk of transmission associated with key interventions (chest compressions, defibrillation, cardiopulmonary resuscitation) to inform international treatment recommendations.	We did not find any direct evidence as to whether chest compressions or defibrillation are associated with aerosol generation or transmission of infection. Data from manikin studies indicates that the donning of personal protective equipment delays treatment delivery. Studies provided only indirect evidence, with no studies describing patients with COVID-19.	It is uncertain whether chest compressions or defibrillation cause aerosol generation or transmission of COVID-19 to rescuers. There is very limited evidence and a rapid need for further studies.
Shadarevian, 2024 [36]	cross-sectional analysis	Understanding the impacts of COVID-19 on bystanders’ willingness to administer CPR in three Canadian provinces.	Participants reported less willingness to perform chest compressions on strangers during the pandemic compared to their recollections before the pandemic. With PPE available, particularly masks, willingness recovered to 91.3%. Willingness varied according to regions. Reluctance to assist older adults increased from 6.6% to 12.0%.	This study highlights changes in CPR willingness during the COVID-19 pandemic, underscoring the importance of PPE and offering insights into public health strategies pertaining to CPR during a pandemic.
Shrimpton, 2024 [37]	prospective study	To quantify the risk of respiratory aerosol generation during CPR in humans.	CPR in humans generates high concentrations of respiratory aerosol and these concentrations were consistently higher than those seen in previous studies of awake and anaesthetized humans (by up to 100-fold), even when coughing or undertaking forced expiratory activities.	CPR generates very high concentrations of respiratory aerosol, potentially increasing the risk of airborne disease transmission. Airborne transmission precautions are recommended during CPR in the setting of high-risk pathogens. Risk is reduced once the airway is secured and connected to a breathing system with a filter.

Abbreviations: AED—automated external defibrillator; bCPR—bystander cardiopulmonary resuscitation; COVID-19—Coronavirus-19; CPR—cardiopulmonary resuscitation; IHCA—in-hospital cardiac arrest; EMS—emergency medical service; OHCA—out-of-hospital cardiac arrest; PPE—personal protective equipment; ROSC—recovery of spontaneous circulation.

## Data Availability

Data are available from the corresponding authors on any acceptable reason.

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
