# Peer review of "Enhancing Safety and Quality of Cardiopulmonary Resuscitation During Coronavirus Pandemic"

_jcm, 2025, doi:10.3390/jcm14124145_

Round 1

Reviewer 1 Report

Comments and Suggestions for Authors

The review "Enhancing Safety and Quality of Cardiopulmonary Resuscitation During the Coronavirus Pandemic" addresses a niche topic within emergency medicine, specifically resuscitation during the COVID-19 period. Below are several suggestions for improvement:

  1. The introduction is too long and contains information that would be more appropriate in the Discussion section. Additionally, statistical significances should be removed from this part of the manuscript.
  2. At the end of the introduction, the aim of the study and its nature (e.g., narrative review or systematic review) should be clearly stated.
  3. Section 2 should be revised to align with the structure of a systematic review, since this format has been adopted. It is recommended to add a PRISMA flow chart and provide details on the selection process, inclusion/exclusion criteria, and search strategy.
  4. The Results section should include more recent articles on this topic. For example, the following article is highly relevant and brings important updates, yet it was not included in the review: 10.3390/jcm13237399.
  5. A summary table compiling the main results and findings of the reviewed studies should be added to offer a structured and comparative overview of the current literature.
  6. The Discussion should be further developed by addressing unclear or controversial aspects from the literature, highlighting areas where further research is needed.
  7. The conclusion section should be more concise and focused.

Author Response

Comment: The review "Enhancing Safety and Quality of Cardiopulmonary Resuscitation During the Coronavirus Pandemic" addresses a niche topic within emergency medicine, specifically resuscitation during the COVID-19 period. Below are several suggestions for improvement:

Response: Thank You for evaluating our manuscript! Explaining the „niche” modality of this manuscript we put the aim of the study in the context of bioethical and efficacy concern of CPR during coronavirus pandemic.New text is signed with blue coloured characters. Please refer lines 81-85!

Comment: 1. The introduction is too long and contains information that would be more appropriate in the Discussion section. Additionally, statistical significances should be removed from this part of the manuscript.

Response: Thank You for the valuable advice! We have decreased the extension of the Introduction and at the same time more comprehensive background was presented citing recent references from the last three years. We have made a new part for epidemiology, statistical significances were removed there at the same time and some parts were replaced to Discussion according to Yor advice.Please especially refer lines 57-75!

Comment: 2. At the end of the introduction, the aim of the study and its nature (e.g., narrative review or systematic review) should be clearly stated.

Response: Thank You for the notice! At the end of the Introduction the aim of the study and its nature (narrative review) is stated. Please refer lines 83-85!

Comment: 3. Section 2 should be revised to align with the structure of a systematic review, since this format has been adopted. It is recommended to add a PRISMA flow chart and provide details on the selection process, inclusion/exclusion criteria, and search strategy.

Response: Thank You for the notice! We have changed the format according to the Editor’s advice (Limitations, Future directioms) fitting for narrative review format. Since this is a narrative review we have not applied the PRISMA flow chart, that is why we can not provide it. We are providing search strategy. Please refer lines 87-92!

Comment: 4. The Results section should include more recent articles on this topic. For example, the following article is highly relevant and brings important updates, yet it was not included in the review: 10.3390/jcm13237399.

Response: Thank You for the advice! More recent articles were critically analyzed and cited among Results. Also a new part on epidemiology (3.1.) was introduced replacing from Introduction. Both bioethical (3.2.) and Efficacy part (3.3.) were updated with more recent data originating from references including Your proposed article as well (Ref.21.) and References were amended with these citations to provide more comprehensive picture. New text is signed with blue coloured characters Please refer lines 126-129, 166-186, 230-262, 270-296, and 312-323!

Comment: 5. A summary table compiling the main results and findings of the reviewed studies should be added to offer a structured and comparative overview of the current literature.

Response: Thank You for Your valuable advice! Two summary tables were added according to the main topics of the manuscript (DNACPR, and proper CPR) presenting the author, year, aim, main results and conclusions of the included studies. Please refer Table 1. and Table 2.!

Comment: 6. The Discussion should be further developed by addressing unclear or controversial aspects from the literature, highlighting areas where further research is needed.

Response: Thank You for the valuable advice! Discussion was developed addressing controversial aspcts. according to the Editor’s advice new Future Directions part were prepared higlighting the unclear aspects where further research is needed.  Please refer lines 355-356, 372-385, and 393-405!

Comment: 7. The conclusion section should be more concise and focused.

Response: Thank You for the suggestion! Conclusions part was corrected with a more concise and focused text. Please refer lines 387-391.

Reviewer 2 Report

Comments and Suggestions for Authors

I have read the review on the impact of the COVID-19 pandemic on CPR and guidelines. The content is highly insightful and includes crucial findings that contribute to future medical practice and research. Below, I present some comments to further enhance the quality of the review.

  1. The data showing that cardiac arrests in public places were halved is highly noteworthy. While the existing discussion attributing this phenomenon to lockdowns, reduced public presence, or other factors is reasonable, further exploration could provide a clearer picture of how the pandemic influenced social structures, behavioral patterns, and ultimately, medical outcomes. For instance, discussing the degree to which each factor contributed, citing related studies, and incorporating data-driven estimations would add depth and persuasiveness to the analysis.
  2. The finding that the survival-to-discharge rate decreased by 50% during the pandemic suggests significant effects on healthcare delivery and CPR quality. To clarify its implications, deeper analysis is necessary. How did delays in the arrival of emergency medical services and postponements in defibrillation and adrenaline administration specifically impact survival rates? How did the strain on medical resources—ICU beds, healthcare personnel—affect post-CPR intensive care and recovery? To what extent did the complexity of PPE usage and concerns over aerosol generation slow down and affect the quality of resuscitation efforts? A more detailed exploration of these interconnected factors would better convey the significance of this finding.
  3. During the pandemic, concerns about aerosol generation in resuscitation were emphasized. If many members of the public hesitated to perform CPR due to infection fears, this likely contributed to lower patient survival rates. It is important to clarify how much of a real risk this concern posed, including psychological barriers to performing CPR. How frequently did civilians who performed resuscitation contract COVID-19? Incorporating available evidence on this would be extremely valuable for future guideline formulation and public awareness efforts.
  4. The current conclusion aims to comprehensively cover key points from the review, but it appears somewhat lengthy. A conclusion should succinctly convey the most critical findings and their overall implications. Some content currently included in the conclusion may be better placed in the "Discussion" section to strengthen connections with the results and allow for a more thorough exploration.

Author Response

Comment: I have read the review on the impact of the COVID-19 pandemic on CPR and guidelines. The content is highly insightful and includes crucial findings that contribute to future medical practice and research. Below, I present some comments to further enhance the quality of the review.

Response: Thank You for Your appreciation! We have amended the thext with the results of critically analized and also cited more recent studies in order to improve the quality and comrehesiveness. We have also changed the format according to the Editor’s instructions (insert Future Directions and Limitation part) fitting for narrative review format. More recent articles were critically analyzed and cited in Relevant Section. Also a new part on epidemiology (3.1.) was introduced replacing from Introduction. Both bioethical (3.2.) and Efficacy part (3.3.) were updated according to more recent cites  We have also created two tables in order to perform detailed informations about reviewed studies. The new text is signed with blue coloured characters.

Comment: 1. The data showing that cardiac arrests in public places were halved is highly noteworthy. While the existing discussion attributing this phenomenon to lockdowns, reduced public presence, or other factors is reasonable, further exploration could provide a clearer picture of how the pandemic influenced social structures, behavioral patterns, and ultimately, medical outcomes. For instance, discussing the degree to which each factor contributed, citing related studies, and incorporating data-driven estimations would add depth and persuasiveness to the analysis.

Response: Thank You for Your valuable advice! We have amended the manuscript with the proposed data (and concerning References as well) concerning social structures, behavioral paterns and medical outcomes to provide more comprehensive picture about the influence of the pandemic in order to enhance breadth and persuasiveness. Please refer lines 230-262!

Comment: 2. The finding that the survival-to-discharge rate decreased by 50% during the pandemic suggests significant effects on healthcare delivery and CPR quality. To clarify its implications, deeper analysis is necessary. How did delays in the arrival of emergency medical services and postponements in defibrillation and adrenaline administration specifically impact survival rates? How did the strain on medical resources—ICU beds, healthcare personnel—affect post-CPR intensive care and recovery? To what extent did the complexity of PPE usage and concerns over aerosol generation slow down and affect the quality of resuscitation efforts? A more detailed exploration of these interconnected factors would better convey the significance of this finding.

Response: Thank You for Your questions and avice! We tried to answer all of Yor questions amending the manuscript with recent clinical data (and concerning References as well) concerning the effect of the delay of EMS, defibrillations, adrenaline and other drug administration and other key factors which all have significant impact on the survival during coronavirus pandemic. The decelerative effect of the strain on medical resources. the PPE usage and concerns over aerosol generations on resuscitation efforts were also introduced analizing the interconnected factors as well. Please refer lines 270-296!

Comment: 3. During the pandemic, concerns about aerosol generation in resuscitation were emphasized. If many members of the public hesitated to perform CPR due to infection fears, this likely contributed to lower patient survival rates. It is important to clarify how much of a real risk this concern posed, including psychological barriers to performing CPR. How frequently did civilians who performed resuscitation contract COVID-19? Incorporating available evidence on this would be extremely valuable for future guideline formulation and public awareness efforts.

Response: Thank You for Your valuable advice and question! We have amended the manuscript with the suggested part including fear and hesitation and other influencing factors of the providers. Very important question is the risk of providers infection during CPR concerning aerosol generating procedures. It is still a controversial question and can be addressed to further prospective studies. Please refer lines 249-262, 291-296, and 312-323!

Comment: 4. The current conclusion aims to comprehensively cover key points from the review, but it appears somewhat lengthy. A conclusion should succinctly convey the most critical findings and their overall implications. Some content currently included in the conclusion may be better placed in the "Discussion" section to strengthen connections with the results and allow for a more thorough exploration.

Response: Thank You for the valuable advice! Discussion was developed covering the most critical findings and their overall implications. Conclusions part was compressed with more thorough and focused text conveying the most critical findings. According to the Editor’s advice new Future Directions part was prepared as well higlighting the unclear aspects where further research is needed. Please refer lines 355-356, 372-385, and 393-405!

Round 2

Reviewer 1 Report

Comments and Suggestions for Authors

The authors have made the requested changes.

Reviewer 2 Report

Comments and Suggestions for Authors

This paper has been appropriately revised according to my instructions.

I believe it has become a highly meaningful study.